# Role-aware Multi-agent Reinforcement Learning for Coordinated Emergency Traffic Control

**Ming Cheng**
Central South University
Changsha, China
244701028@csu.edu.cn

**Hao Chen**
City University of Macau
Macao, China
sundaychenhao@gmail.com

**Zhiqing Li**
City University of Macau
Macao, China
zhiqinglizzy@gmail.com

**Jia Wang**
Xi'an Jiaotong-Liverpool University
Suzhou, China
jia.wang02@xjtlu.edu.cn

**Senzhang Wang**[*]
Central South University
Changsha, China
szwang@csu.edu.cn

## Abstract

Emergency traffic control presents an increasingly critical challenge, requiring seamless coordination among emergency vehicles, regular vehicles, and traffic lights to ensure efficient passage for all vehicles. Existing models primarily only focus on traffic light control, leaving emergency and regular vehicles prone to delay due to the lack of navigation strategies. To address this issue, we propose the **Role-aware Multi-agent Traffic Control (RMTC)** framework, which dynamically assigns appropriate roles to traffic components for better cooperation by considering their relations with emergency vehicles and adaptively adjusting their policies. Specifically, RMTC introduces a *Heterogeneous Temporal Traffic Graph (HTTG)* to model the spatial and temporal relationships among all traffic components (traffic lights, regular and emergency vehicles) at each time step. Furthermore, we develop a *Dynamic Role Learning* model to infer the evolving roles of traffic lights and regular vehicles based on HTTG. Finally, we present a *Role-aware Multi-agent Reinforcement Learning* approach that learns traffic policies conditioned on the dynamically roles. Extensive experiments across four public traffic scenarios show that RMTC outperforms existing traffic light control methods by significantly reducing emergency vehicle travel time, while effectively preserving traffic efficiency for regular vehicles. The code is released at `https://github.com/mingchenghexi/RMTC`.

## 1 Introduction

Quick arrival of Emergency Vehicles (EMVs), such as fire trucks, ambulances, and police cars, is crucial in modern emergencies, including medical crises [1], fires [2], and accidents [3]. However, managing emergency traffic response is inherently complex, as EMVs not only impact the travel time of regular vehicles (REVs), but the congestion caused by REVs also affects the arrival time of EMVs. As illustrated in Figure 1, when traffic congestion occurs, traditional traffic light control often routes emergency vehicles (EMVs) along the shortest path. In contrast, by integrating vehicle navigation, EMVs can be guided to bypass congested segments, while surrounding REVs are steered away from the paths that EMVs will traverse. This coordination alleviates conflicts and improves overall traffic efficiency. In this context, the challenge lies in how to navigate both emergency and regular vehicles

---

[*]Corresponding author.

39th Conference on Neural Information Processing Systems (NeurIPS 2025).

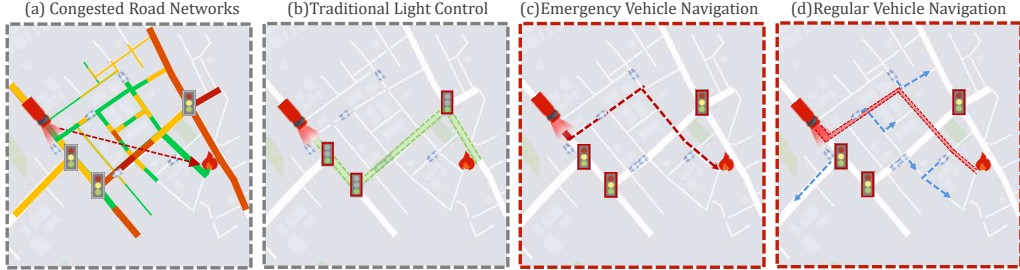

Figure 1: Visualization of (a) Road Network Congestion, (b) EMV Routes under Traditional light Control, (c) EMV Routing with Navigation, and (d) REV Routing with Navigation

while coordinating with traffic lights to achieve efficient traffic flow for both EMVs and REVs. This research can be integrated into navigation software, such as Google Maps and Amap, to help save property, lives, and other critical resources in real-world scenarios.

Existing models mainly focus on traffic light control and can be broadly classified into two categories: rule-based methods and reinforcement learning (RL)-based methods. Rule-based methods usually give EMVs higher priority like more green lights to accelerate EMV movement. Greedy approaches [4, 5] switch the corresponding traffic signal to green while halting conflicting directions to facilitate EMV passage. RL-based models, on the other hand, use the arrival time of EMVs as an optimization objective to adjust traffic light policies. For example, EMVLight [6] assigns distinct rewards to different traffic signals to encourage coordinated responses to EMVs, while RECAL [7] extends traffic control to scenarios involving multiple EMVs. However, these existing methods focus solely on controlling traffic lights and fail to coordinate the behavior of traffic lights and vehicles.

Though considering EMVs, REVs, and traffic lights is an attractive approach, but it introduces the following three key challenges when incorporating three types of traffic components: EMVs, REVs, and traffic lights. **(1) Personalized behavior within the same type of components:** Traffic lights and vehicles in different positions should make distinct decisions. For instance, lights and vehicles near EMVs should provide assistance, while those farther away should operate normally. **(2) Interdependencies between different types of components:** Traffic lights directly influence the movement of vehicles, while EMVs and REVs mutually impact each other's navigation. **(3) The seesaw problem between emergency and regular vehicles:** Balancing the needs of EMVs and REVs is challenging, as facilitating EMVs often comes at the expense of REVs' efficiency.

Motivated by these challenges, we propose the **R**ole-aware **M**ulti-agent **T**raffic **C**ontrol (RMTC) framework, which dynamically considers the relationships among all traffic components, assigns appropriate roles to these components, and adjusts their policies accordingly. Since the relationships between traffic components evolve over time, RMTC introduces a *Heterogeneous Temporal Traffic Graph (HTTG)* to model the spatial and temporal relationships among traffic components at each time step. Furthermore, we develop a *Dynamic Role Learning* model to infer the evolving roles of traffic lights and REVs based on HTTG. Finally, we present a *Role-aware Multi-agent Reinforcement Learning* approach to learn role-aware traffic policies conditioned on the dynamically roles. In summary, our contributions are as follows:

- We explore the novel problem of multi-agent emergency traffic control by defining the optimization problem for traffic lights, EMVs, and REVs. This exploration can help improve real-world navigation software to save more property and lives.

- We design a Heterogeneous Temporal Traffic Graph (HTTG) and a corresponding Dynamic Role Learning model to infer the roles of individual traffic components.

- We design tailored rewards and train a Role-aware Multi-agent Reinforcement Learning model for the coordinated control of traffic lights, EMVs, and REVs.

- Evaluations on four public traffic datasets demonstrate that RMTC consistently outperforms existing traffic light control methods in terms of both EMV travel time and overall traffic efficiency.

## 2  Preliminaries

**Notations**  We define a multi-agent traffic control consisting of three types of agents: EMVs agents $\mathcal{I}_{\text{emv}}$, REVs agents $\mathcal{I}_{\text{rev}}$, and traffic light agents $\mathcal{I}_{\text{tl}}$. Each agent functions within a dynamic traffic network where its actions influence and are influenced by other agents.

At each discrete time step $t$, an agent receives an observation $o^{(t)}$ that includes local traffic state. For a traffic light agent, the observation includes the current signal phase, the number of approaching vehicles, the queue length of stopped vehicles, lane occupancy, vehicle flow rate, the number of fully stopped vehicles, and traffic pressure. For a vehicle agent, the observation includes the current intersection, current lane, destination intersection, and destination lane.

Based on its observation, each agent selects an action $a^{(t)}$ from its predefined action space. For traffic light agents, the action corresponds to selecting the next traffic phase from a predefined set of non-conflicting options. When a phase is activated, only vehicles on the associated lanes are allowed to move. For vehicle agents, the action space consists of routing decisions at intersections, including turning left, going straight, or turning right.

The environment subsequently evaluates these actions through extrinsic rewards. For each traffic light agent $i \in \mathcal{I}_{\text{tl}}$, the extrinsic reward is provided by the environment based on key traffic metrics collected from the lanes it controls, including queue length, average waiting time, traffic pressure, and vehicle delay. For each vehicle agent $i \in \mathcal{I}_{\text{rev}} \cup \mathcal{I}_{\text{emv}}$, the extrinsic reward is defined as the negative elapsed travel time since its last decision point, which encourages more efficient routing decisions [8]:

$$r_{\text{extr}}^{(t)} = -T_{\text{elapsed}}^{(t)}, \tag{1}$$

**Problem Definition**  The coordinated control of traffic lights, EMVs navigation, and REVs navigation aims to enable rapid passage of EMVs while maintaining overall traffic flow efficiency. This is formulated as learning a joint multi-agent policy $\{\pi_{\theta_i}\}_{i \in \mathcal{I}}$, parameterized by $\theta_i$, that maximizes the expected cumulative reward across all agents:

$$\max_{\{\pi_{\theta_i}\}} \mathbb{E}\left[\sum_{t=0}^{T} \sum_{i \in \mathcal{I}} \gamma^t r_i^{(t)}\right], \tag{2}$$

where $\mathcal{I} = \mathcal{I}_{\text{emv}} \cup \mathcal{I}_{\text{rev}} \cup \mathcal{I}_{\text{tl}}$, and $\gamma \in (0, 1)$ is the discount factor, and $T$ denotes the episode length.

## 3  Role-aware Multi-agent Controlling

To jointly optimize vehicle navigation and traffic signal control in emergency scenarios, we propose RMTC, a Role-aware Multi-agent Traffic Controlling framework. As illustrated in Figure 2, we construct a heterogeneous temporal traffic graph based on the traffic environment and enable dynamic role learning through three constraints: EMV Position Role Impacting, EMV Trajectory Role Impacting, and Role Consistency Constraint. The learned dynamic role features are then used to compute each agent's state and reward. Through role-aware multi-agent reinforcement learning, agents are guided to make decisions that improve both EMV efficiency and overall traffic flow.

### 3.1  Heterogeneous Temporal Traffic Graph (HTTG)

The Heterogeneous Temporal Traffic Graph (HTTG), denoted as $G^{(t)} = (\mathcal{I}, E^{(t)})$, is used to represent the interactions among diverse traffic agents. The node set $\mathcal{I}$ comprises all agents, while the edge set $E^{(t)}$ evolves over time to reflect dynamic and heterogeneous relationships. Specifically, the HTTG is updated at each step $t$, allowing the graph structure to dynamically reflect real-time relationships among EMVs, REVs, and traffic light.

**HTTG Edge Construction**  The HTTG is designed to capture the interaction relationships among agents. Specifically, the edge set $E^{(t)}$ comprises four relation types, forming the relation set $\mathcal{R}$ used in role-aware message passing.

*1) TL-REV edges:* Each light node $i_{\text{tl}}$ is connected to REVs $i_{\text{rev}}$ currently on its incoming lanes, modeling the regulatory influence of lights.

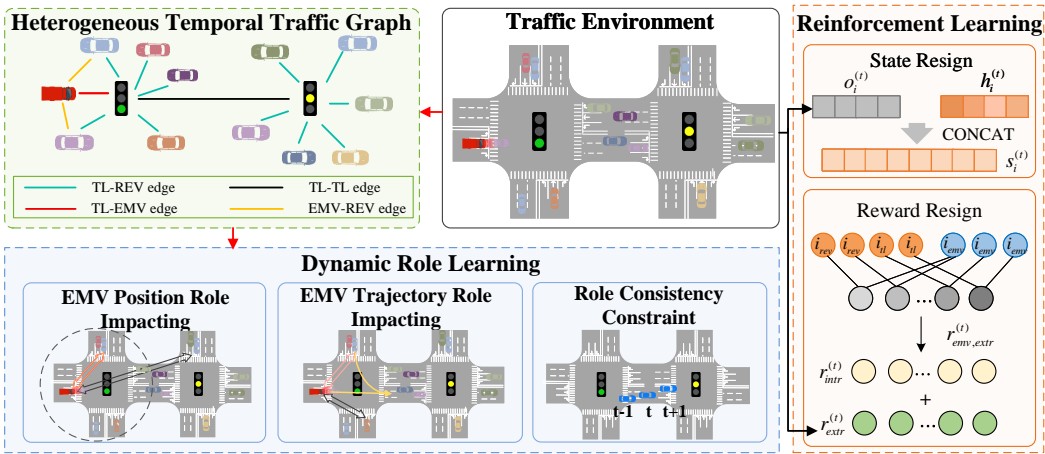

Figure 2: The overview framework of RMTC. RMTC constructs a heterogeneous temporal traffic graph based on the traffic environment, employs dynamic role learning to assign adaptive roles to agents, and leverages these dynamic roles to design states and rewards for reinforcement learning.

*2) TL-EMV edges:* Each light node $i_{\text{tl}}$ connects to EMVs $i_{\text{emv}}$ within its control incoming lanes, enabling lights to be aware of and responsive to approaching EMVs.

*3) TL-TL edges:* Two lights nodes $i_{\text{tl},i}$ and $i_{\text{tl},j}$ are connected if their intersections are adjacent in the road network, capturing coordination needs between neighboring intersections.

*4) EMV-REV edges:* REVs with motion patterns similar to EMVs are more likely to be influenced by their presence, such as adjusting their trajectories. Based on this intuition, EMVs are connected to REVs located on the same inbound lanes controlled by a traffic light, if their observed states are sufficiently similar. We use cosine similarity to measure this relationship, and add an edge when the similarity exceeds a predefined threshold $\delta$:

$$s^{(t)}(i_{\text{emv}}, i_{\text{rev}}) = \cos(\boldsymbol{o}_{i_{\text{emv}}}^{(t)}, \boldsymbol{o}_{i_{\text{rev}}}^{(t)}) = \frac{\boldsymbol{o}_{i_{\text{emv}}}^{(t)} \cdot \boldsymbol{o}_{i_{\text{rev}}}^{(t)}}{\|\boldsymbol{o}_{i_{\text{emv}}}^{(t)}\|\|\boldsymbol{o}_{i_{\text{rev}}}^{(t)}\|}, \tag{3}$$

where $\boldsymbol{o}_{i_{\text{emv}}}^{(t)}$ and $\boldsymbol{o}_{i_{\text{rev}}}^{(t)}$ denote the observed state of the EMVs and REVs, respectively.

**Heterogeneous Role Convolution**   The heterogeneous graph-based role encoder facilitates structural representation learning by capturing the interdependencies among agents and distinguishing their dynamic roles within the traffic environment. Specifically, it employs a message-passing mechanism that iteratively updates each node's representation. This process aggregates and integrates information from neighboring nodes, carefully considering the heterogeneous nature of node and edge types. The message-passing mechanism can be expressed as:

$$\boldsymbol{h}_i^{(t,l)} = \sigma\left(\sum_{r \in \mathcal{R}} \sum_{u \in \mathcal{N}_r^{(t)}(i)} \mathbf{W}_r^{(l)} \cdot \boldsymbol{h}_u^{(t,l-1)} + \boldsymbol{b}_r^{(l)}\right), \tag{4}$$

where $\boldsymbol{h}_i^{(t,l)}$ denotes the role feature of agent $i$ at step $t$ and layer $l$, and is initialized as $\boldsymbol{h}_i^{(t,0)} = \boldsymbol{o}_i^{(t)}$. $\mathcal{N}_r^{(t)}(i)$ denotes the set of neighbors of agent $n$ connected by edge type $r$, where $i \in \{\mathcal{I}_{\text{emv}}, \mathcal{I}_{\text{rev}}, \mathcal{I}_{\text{tl}}\}$. $\mathbf{W}_r^{(l)}$ is the learnable weight matrix for edge type $r$ at layer $l$, $\boldsymbol{b}_r^{(l)}$ is the bias term, and $\sigma(\cdot)$ is a non-linear activation function. The layer index $l$ denotes the propagation depth in the message-passing process, where a two-layer architecture is employed.

## 3.2 Dynamic Role Learning

Dynamic role learning captures role features of traffic agents through HTTG at each time $t$. These role features are influenced by the positions and trajectories of EMVs, leading to changes in the

agents' behavior, such as yielding, accelerating, or changing lanes. At the same time, it is important that these behavioral patterns remain stable in the short term, avoiding rapid fluctuations. Based on these considerations, we design three learning objectives to incorporate the influence of EMVs into role representation learning while ensuring stability over time.

**EMV Position Role Impacting**  The position of the EMV relative to other agents plays a crucial role in shaping their behavior. Agents closer to the EMV are likely to prioritize actions that allow the EMV to pass quickly, such as adjusting traffic signals or changing their movement. In contrast, agents farther away may exhibit different behaviors.

Therefore, we maximize a lower bound of the mutual information between each agent's role embedding $\boldsymbol{h}_n^{(t)}$ and the EMV-related graph features $\boldsymbol{h}_{\text{emv}}^{(t)}$, ensuring role embeddings reflect the EMV's positional impact effectively. The lower bound of the mutual information can be expressed as (the proof is deferred to Appendix B):

$$I^{(t)}(\boldsymbol{h}^{(t)}; \boldsymbol{h}_{\text{emv}}^{(t)}) \geq \frac{1}{N_n^{(t)}} \sum_{i=1}^{N_n^{(t)}} \left[ T_\omega(\boldsymbol{h}_i^{(t)}, \boldsymbol{h}_{\text{emv}}^{(t)}) - \log \left( \frac{1}{N_n^{(t)}} \sum_{j=1}^{N_n^{(t)}} \exp \left( T_\omega(\boldsymbol{h}_j^{(t)}, \boldsymbol{h}_{\text{emv}}^{(t)}) \right) \right) \right], \quad (5)$$

where $\boldsymbol{h}_i^{(t)}$ and $\boldsymbol{h}_{\text{emv}}^{(t)}$ denote the role embedding of the $i$-th agent and the EMVs features in the same graph, respectively. $N_n^{(t)}$ denotes the number of agent $n$ at time $t$, $n \in \{\mathcal{I}_{\text{rev}}, \mathcal{I}_{\text{tl}}\}$. The function $T_\omega(\cdot, \cdot)$ is a learnable critic parameterized by $\omega$.

**EMV Trajectory Role Impacting**  The trajectory of the EMV not only influences its own future decisions but also affects the behavior of other agents in the current time step. For example, if the EMV is approaching a light-controlled intersection, the light controllers at that intersection must anticipate the EMV's next decision (e.g., whether it will go straight or turn) and adjust the traffic light phases accordingly, even before the EMV arrives. Similarly, agents further along the EMV's path must adjust their actions based on the EMV's predicted trajectory to maintain smooth flow. The trajectory information of the EMV must be maximized within the role representations.

Specifically, we encode the EMV's observation history using an LSTM network and optimize the following mutual information lower bound:

$$I^{(t)}(\boldsymbol{h}^{(t)}; \boldsymbol{g}_{\text{emv}}^{(t)}) \geq \frac{1}{N_n^{(t)}} \sum_{i=1}^{N_n^{(t)}} \log \left( \frac{\exp \left( T_{\omega'}(\boldsymbol{h}_i^{(t)}, \boldsymbol{g}_{\text{emv}}^{(t)}) \right)}{\sum_{j=1}^{N_n^{(t)}} \exp \left( T_{\omega'}(\boldsymbol{h}_j^{(t)}, \boldsymbol{g}_{\text{emv}}^{(t)}) \right)} \right), \quad (6)$$

$$\boldsymbol{g}_{\text{emv}}^{(t)} = \text{GRU} \left( \boldsymbol{o}_{\text{emv}}^{(t)}, \boldsymbol{g}_{\text{emv}}^{(t-1)} \right), \quad (7)$$

where $\boldsymbol{h}_i^{(t)}$ denotes the role embedding of the $i$-th agent, and $\boldsymbol{g}_{\text{emv}}^{(t)}$ is the hidden state of the GRU module at time step $t$. The function $T_{\omega'}(\cdot, \cdot)$ is a learnable critic, parameterized by $\omega'$.

**Role Consistency Constraint**  Role consistency over short time intervals is essential for stable decision-making. If role features rely only on local observations, they may fluctuate significantly due to transient changes in the environment, leading to unstable or suboptimal behaviors.

Therefore, we promote temporal consistency in roles by smoothing their evolution over time. Specifically, we apply a contrastive objective that draws role embeddings from adjacent time steps closer while pushing them apart from those of unrelated agents or distant moments. This objective is formulated as:

$$\mathcal{L}_{\text{cons}}^{(t)} = -\sum_{i=1}^{N_n^{(t)}} \log \left( \frac{\exp \left( d(\boldsymbol{h}_i^{(t)}, \boldsymbol{h}_i^{(t+1)})/\tau \right)}{\sum_{k \in N_i^{(t)}} \exp \left( d(\boldsymbol{h}_i^{(t)}, \boldsymbol{h}_k^{(t+1)})/\tau \right)} \right), \quad (8)$$

where $d(\cdot)$ is a distance metric incorporating behavior-relevant state changes, and $\tau$ is a fixed temperature parameter. Therefore, the overall training loss can be formulated as:

$$\mathcal{L}_{\text{role}} = \sum_{t=1}^{T} \left( -\left( I^{(t)}(\boldsymbol{h}^{(t)}; \boldsymbol{h}_{\text{emv}}^{(t)}) + I(\boldsymbol{h}^{(t)}; \boldsymbol{g}_{\text{emv}}^{(t)}) \right) + \mathcal{L}_{\text{cons}}^{(t)} \right), \quad (9)$$

where the MI terms ensure EMV-aware adaptability, and the temporal consistency term $\mathcal{L}_{\text{cons}}^{(t)}$ maintains smooth role transitions over consecutive time steps, together enabling the embeddings to be both responsive to dynamic changes and stable enough for effective decision-making.

## 3.3 Role-aware Multi-Agent Reinforcement Learning

Based on the learned dynamic role, we develop a multi-agent reinforcement learning framework for traffic control, which consists of four components:

**Policy Designing** The state of agent is formed by concatenating its learned role feature embedding $\boldsymbol{h}^{(t)}$ , which captures the agent's dynamic role, with its environmental observations $\boldsymbol{o}^{(t)}$. The concatenated state, integrating both environmental features and role information, is then input to the actor network, which maps the input to a distribution over actions:

$$\pi_\theta(a^{(t)}|\boldsymbol{o}^{(t)}, \boldsymbol{h}^{(t)}) = f_\theta\big(\text{concat}(\boldsymbol{o}^{(t)}, \boldsymbol{h}^{(t)})\big) \tag{10}$$

where $\boldsymbol{o}^{(t)}$ denotes the agent's observations at time $t$, $\boldsymbol{h}^{(t)}$ represents the learned role embedding, and $f_\theta$ is the policy network with parameters $\theta$ mapping concatenated inputs to actions.

**Role-aware Intrinsic Reward** Role embeddings capture behavior patterns of agents, but they require feedback to guide role-consistent actions. We propose a role-aware intrinsic reward mechanism that encourages agents to align with their roles under EMVs influence.

Specifically, we quantify the influence of EMVs on each agent by computing the cosine similarity between the agent's role embedding $\boldsymbol{h}_n^{(t)}$ and the EMVs feature vector $\boldsymbol{h}_{\text{emv}}^{(t)}$. Using these similarity scores, we select the top-$k$ agents whose role embeddings are most relevant to the current EMVs context. The intrinsic reward for each agent is then computed as a weighted version of the EMVs' reward, where the weight is determined by the similarity score between the agent's role and the EMVs. Therefore, the intrinsic reward for each agent can be expressed as:

$$r_{n,\text{intr}}^{(t)} = \sum_{i=1}^{N_{emv}} s\left(\boldsymbol{h}_n^{(t)}, \boldsymbol{h}_{\text{emv},m}^{(t)}\right) \cdot r_{\text{emv},i,\text{extr}}^{(t)}, \quad \forall n \in \{I_{\text{rev}}, I_{\text{tl}}\}, \tag{11}$$

where, $r_{\text{emv},i,\text{extr}}^{(t)}$ denotes the extrinsic reward received by EMV i.

Finally, this role-aware intrinsic reward is combined with the agent's extrinsic reward $r_{\text{extr}}^{(t)}$ during training to form the total reward:

$$r_n^{(t)} = r_{n,\text{extr}}^{(t)} + \lambda \cdot r_{n,\text{intr}}^{(t)}, \tag{12}$$

where $\lambda$ is a scaling factor adjusted according to the magnitude of agent extrinsic reward.

**Multi-Agent Optimization** We adopt Proximal Policy Optimization (PPO), a widely used policy-based reinforcement learning algorithm that optimizes a policy by maximizing expected cumulative rewards. In this multi-agent setting, each agent, including vehicles and traffic lights, learns its own policy based on local observations and role-specific features.

To improve scalability and coordination efficiency, a shared policy network is adopted across agents with similar roles (e.g., vehicles or traffic lights). This shared-parameter strategy allows agents to learn from diverse experiences while significantly reducing the computational burden of maintaining a separate policy for each agent. The training objective includes both policy and value updates, and the loss is defined as follows:

$$\mathcal{L}(\theta, \phi) = \mathbb{E}\left[ \min\left(\delta^{(t)}(\theta)\,\mathbf{A}^{(t)}, \text{clip}\left(\delta^{(t)}(\theta), 1-\epsilon, 1+\epsilon\right)\mathbf{A}^{(t)}\right) + \left(V_\phi(\boldsymbol{o}^{(t)}, \boldsymbol{h}^{(t)}) - R^{(t)}\right)^2 \right], \tag{13}$$

$$\mathbf{A}^{(t)} = \sum_{l=0}^{T} (\gamma\psi)^l \left( r_{\text{extr}}^{(t)} + \lambda r_{\text{intr}}^{(t)}(\boldsymbol{h}^{(t)}) + \gamma V_\phi(\boldsymbol{o}^{(t+1)}, \boldsymbol{h}^{(t+1)}) - V_\phi(\boldsymbol{o}^{(t)}, \boldsymbol{h}^{(t)}) \right), \tag{14}$$

Table 1: Comparison of EMV and REV Travel Times for RMTC and Baselines Across Four Datasets. ** indicates the statistical significance p < 0.01 compared to the best-performed baseline, * indicates the statistical significance p < 0.05 compared to the best-performed baseline. "Improve" denotes the percentage reduction in travel time relative to the best baseline.

| Datasets | Synthetic | | | | Real-world | | | |
| --- | --- | --- | --- | --- | --- | --- | --- | --- |
| | Grid 4×4 | | Avenue 4×4 | | Cologne8 | | FengLin | |
| Metrics | $T_{\text{emv}}$ | $T_{\text{rev}}$ | $T_{\text{emv}}$ | $T_{\text{rev}}$ | $T_{\text{emv}}$ | $T_{\text{rev}}$ | $T_{\text{emv}}$ | $T_{\text{rev}}$ |
| FixedTime | 158.00 | 209.12 | 237.00 | 666.90 | 48.00 | 114.96 | 98.00 | 262.74 |
| MaxPressure | 162.00 | 218.45 | 155.00 | 768.82 | 48.00 | 184.67 | 136.00 | 304.24 |
| CoLight | 170.00 | 239.93 | 262.00 | 520.71 | 52.00 | 98.53 | 55.00 | 307.12 |
| IPPO | 143.00 | 171.20 | 294.00 | 574.89 | 75.00 | 96.55 | 48.00 | 230.91 |
| rMAPPO | 118.00 | 171.44 | 250.00 | 519.92 | 53.00 | 95.38 | 35.00 | 398.76 |
| X-Light | 117.00 | 170.37 | 170.00 | 587.02 | 49.00 | 96.96 | 48.00 | 234.90 |
| RECAL | 129.00 | 296.39 | 162.00 | 757.83 | 47.00 | 159.93 | 38.00 | 276.91 |
| EMVlight | 117.00 | 176.03 | 254.00 | 625.85 | 47.00 | 148.9 | 34.00 | 293.79 |
| RMTC | **112.00**** | **163.62**** | **141.00**** | **491.49**** | **36.00**** | **87.90**** | **26.00**** | **218.91**** |
| Improve (%) | 4.27% | 3.96% | 9.03% | 5.46% | 23.40% | 7.84% | 23.52% | 5.19% |

where $\delta^{(t)}(\theta) = \frac{\pi_\theta(a^{(t)}|\boldsymbol{o}^{(t)}\boldsymbol{h}^{(t)})}{\pi_{\theta_{\text{old}}}(a^{(t)}|\boldsymbol{o}^{(t)},\boldsymbol{h}^{(t)})}$ is the importance sampling ratio, $V_\phi$ is the value function approximator with parameters $\phi$, and $R^{(t)}$ is the cumulative return at time $t$. Additionally, $\mathbf{A}^{(t)}$ is the role-aware advantage function incorporating both extrinsic reward $r_{\text{extr}}^{(t)}$ and intrinsic reward $r_{\text{intr}}^{(t)}$, with $\gamma$ as the discount factor, $\psi$ as the decay factor for multi-step advantage estimation.

## 4 Experiments

### 4.1 Experimental Setup

**Experiment Settings.** We conduct our experiments using SUMO[2], a widely adopted traffic simulator that provides control over vehicle movements and traffic signals through its TraCI API. Following common settings in prior work [9], we configure each intersection with eight signal phases to ensure stable signal transitions.

**Datasets.** Our datasets include two synthetic traffic scenarios, Grid4×4 and Avenue, as well as two real-world scenarios, Cologne8 and FengLin [10, 11]. The synthetic scenarios are constructed with idealized grid layouts containing uniformly distributed four-arm intersections. The real-world scenarios feature a mix of two-arm, three-arm, and four-arm intersections.

In our framework, each vehicle and each traffic signal is modeled as an individual agent. The total number of vehicles in each episode is determined by the vehicle generation rate and the duration of the traffic event, allowing us to simulate varying traffic densities. To simulate realistic emergency conditions, we schedule the EMVs to depart from the middle point of the traffic episode. Details about these datasets are shown in Appendix C.1.

**Baselines.** We compare our method with a range of widely used traffic signal control approaches, including both rule-based and reinforcement learning (RL)-based methods. The rule-based baselines include Fixed-Time (FT) [12] and Max-Pressure (MP) [13], while the RL-based baselines include CoLight [14], IPPO [15], rMAPPO [16], X-Light [9], RECAL [7], and EMVlight [6].

To adapt these methods to emergency scenarios, we adopt a standard signal preemption strategy that extends the green phase along the EMV's route, creating a green wave to prioritize its passage [17]. We apply this green wave mechanism to each baseline. Further implementation details are provided in Appendix C.2.

---

[2]https://www.eclipse.org/sumo/

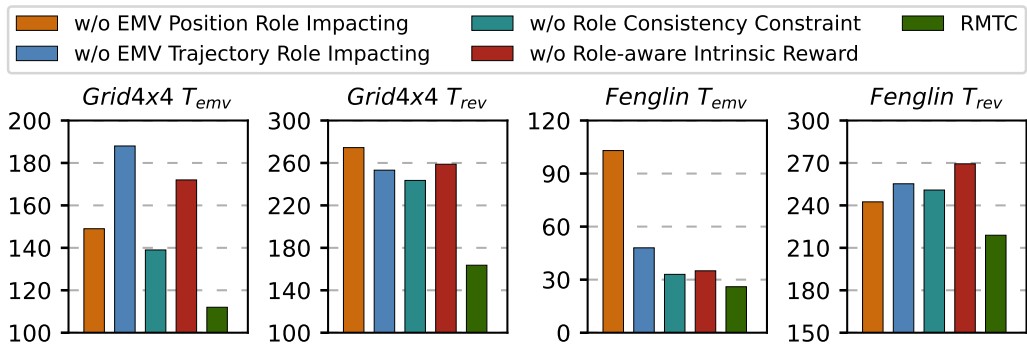

Figure 3: Ablation study results between RMTC with its four variants on Grid 4×4.

**Evaluation Metrics** Following prior studies [6, 9], we adopt the average travel time of both EMVs and REVs to assess overall traffic efficiency and emergency response performance. The average travel time of REVs is defined as:

$$T_{\text{rev}} = \frac{1}{N} \sum_{i=1}^{N} (t_i^{\text{exit}} - t_i^{\text{entry}}),  \tag{15}$$

where $N$ denotes the total number of REVs, and $t_i^{\text{entry}}$, $t_i^{\text{exit}}$ are the entry and exit times of vehicle $i$. The average travel time of EMVs $T_{\text{emv}}$ is computed in the same manner.

### 4.2  Main Results Analysis

The main comparison results for both EMVs and REVs are presented in Table 1. From the results, we can derive the following observations. RMTC achieves a favorable balance by ensuring the fast arrival of EMVs while maintaining overall traffic efficiency.

**Overall Performance Analysis.** Compared to the best-performing baseline methods significantly reduces EMV travel times across all datasets. Notably, it achieves over 15% reduction on three benchmarks: Cologne8 (23.40%) and FengLin (23.52%). Meanwhile, RMTC also improves REV travel times by 7.84% and 5.19% on the same datasets, indicating that prioritizing EMV passage does not come at the cost of increased congestion for regular traffic.

**Comparison with Rule-Based Methods.** Rule-based methods with pre-emption strategies can improve EMV travel time but often disrupt regular traffic, resulting in increased delays. Their reliance on fixed strategies limits adaptability to dynamic traffic flow variations. Consequently, these methods tend to prolong green phases for EMV without accounting for downstream congestion, which exacerbates delays for regular vehicles. In contrast, RMTC dynamically coordinates signal control and vehicle decisions based on real-time traffic states, enabling responsive and balanced traffic management.

**Comparison with RL-Based Methods.** Compared to existing RL-based approaches, RMTC achieves superior EMV response efficiency and lower average travel times for all vehicles across diverse traffic conditions. Traditional RL-based traffic signal control methods, such as CoLight and PressLight, focus primarily on ordinary traffic flows and do not explicitly account for emergency vehicles, limiting their responsiveness when EMVs appear and making it difficult to capture the dynamic traffic changes induced by EMV movements. RL-based methods tailored for emergency scenarios, including RECAL and EMVlight, effectively prioritize EMV clearance but offer limited benefits for ordinary vehicles, which can lead to suboptimal overall traffic performance. In contrast, RMTC jointly learns both vehicle and signal policies, enabling adaptive coordination in dynamic traffic environments. This joint optimization ensures rapid EMV passage while substantially improving the efficiency of ordinary vehicles, resulting in balanced and globally enhanced traffic flow.

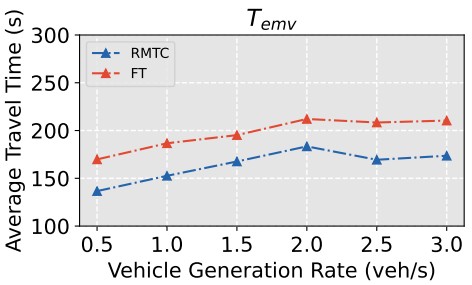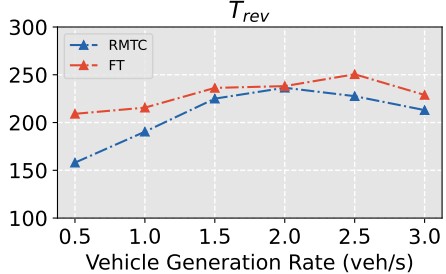

Figure 4: Performance comparison between FT and RMTC under varying congestion on Grid 4×4.

### 4.3 Ablation Study

To evaluate the contributions of each component in our model, we compare the full model with four ablated variants: (i) *w/o EMV Position Role Impacting* removes the constraint of EMV position; (ii) *w/o EMV Trajectory Role Impacting* removes the constraint of EMV action trajectory; (iii) *w/o Role Consistency Constraint* removes the temporal consistency constraint; (iv) *w/o Role-aware Intrinsic Reward* removes the role-aware intrinsic reward, using only the environment reward. From the results shown in Figure 3, we draw the following observations.

**Effectiveness of components in the dynamic role loss.** The performance degradation in *w/o EMV Position Role Impacting* and *w/o EMV Trajectory Role Impacting* demonstrates the effectiveness of enabling agents to learn roles by observing the position and behavioral tendencies of the EMVs. This alignment enables agents to learn role features that support coordination, allowing EMVs to pass through intersections more quickly. The inferior performance of *w/o Role Consistency Constraint* indicates that preventing abrupt shifts in role assignment across time steps is crucial. Without proper scheduling, the learned roles become volatile, which affects overall performance.

**Effectiveness of components in the reward function.** The poor performance of *w/o Role-aware Intrinsic Reward* indicates that, without a reward signal, agents lack proper incentives to prioritize actions that facilitate EMVs passage and minimize delays for other vehicles. Without a proper guidance, agents cannot coordinate effectively, failing to support the rapid movement of EMVs and maintain efficient overall traffic flow.

### 4.4 Performance Comparision under Different Traffic Flows

Figure 4 illustrates the performance of RMTC under different traffic generation rates, ranging from 1 to 3 vehicles per second, where the generation rate reflects the average number of vehicles entering the network per second and thus controls the overall traffic density. Across all rates, RMTC consistently achieves lower average travel times compared to the FT method, and its performance remains stable as traffic load increases, showing no significant degradation. These results confirm that the role-aware embeddings and reward redistribution mechanism enable RMTC to adaptively coordinate agents in real time, effectively prioritizing EMVs while preserving smooth traffic flow for ordinary vehicles even under dynamic and high-density conditions.

### 4.5 Impact of EMV Quantity on Traffic Efficiency

RMTC encodes the relationships between each agent and multiple EMVs using a heterogeneous graph and learns role embeddings via mutual information maximization. This design allows agent roles to reflect varying importance to different EMVs based on their spatial positions. For example, an agent closer to EMV A than EMV B will have a role embedding that more strongly captures EMV A's features, guiding its behavior accordingly. In addition, RMTC incorporates a role-similarity-based reward redistribution mechanism, which enables soft prioritization when multiple EMVs coexist, ensuring coordinated decision-making without severely disrupting regular traffic.

We compare RMTC with IPPO and rMAPPO on the Grid dataset under varying numbers of EMVs (from 1 to 5), as summarized in Table 2. As the number of EMVs increases, the average travel time of REVs in IPPO and rMAPPO rises sharply due to growing route conflicts and the lack of explicit

Table 2: Comparison of travel times under varying numbers of EMVs on the Grid 4×4.

| EMV Travel Time | | | | | |
|---|---|---|---|---|---|
| Number of EMVs | 1 | 2 | 3 | 4 | 5 |
| IPPO | 143.00 | 124.00 | 141.33 | 145.25 | 149.40 |
| rMAPPO | 118.00 | 132.00 | 149.00 | 160.75 | 141.20 |
| RMTC | **112.00** | **108.50** | **125.00** | **133.50** | **136.60** |
| REV Travel Time | | | | | |
| Number of EMVs | 1 | 2 | 3 | 4 | 5 |
| IPPO | 171.20 | 183.81 | 181.22 | 173.87 | 190.59 |
| rMAPPO | 171.44 | 185.26 | 173.08 | 172.37 | 179.04 |
| RMTC | **163.62** | **159.73** | **167.60** | **160.95** | **169.78** |

coordination among agents. In contrast, RMTC maintains relatively stable REV travel times even in dense EMV scenarios, benefiting from its ability to dynamically adapt agent behaviors through learned roles and coordinated signal-phase adjustments.

For EMVs, RMTC consistently outperforms the baselines across all settings, achieving significantly shorter travel times by leveraging structured role modeling, trajectory-aware coordination, and proactive traffic light control. This demonstrates that RMTC not only prioritizes emergency response but also preserves overall traffic efficiency. These results collectively highlight RMTC's robustness and scalability in handling complex multi-EMV environments while maintaining balanced performance for both emergency and regular vehicles.

Additional experimental results for RMTC are provided in the Appendix F.

## 5 Conclusion

Ensuring the rapid passage of EMVs in critical scenarios is essential for minimizing losses and saving lives. However, existing methods primarily focus on traffic light control while overlooking the collaborative between EMVs, REVs, and traffic signals.To address this issue, We propose the Role-aware Multi-agent Traffic Control (RMTC) framework, which jointly coordinates traffic lights and vehicle behaviors through dynamic role modeling. By constructing a Temporal Heterogeneous Traffic Graph and learning adaptive roles for each agent, RMTC enables fine-grained policy adaptation based on evolving traffic contexts. Extensive experiments demonstrate that RMTC significantly accelerates EMVs arrivals while maintaining overall traffic efficiency.

## Acknowledgement

This research was funded by the National Science Foundation of China (No. 62572489, No. 62502008), the Funding Scheme for Research and Innovation of FDCT (Grant No. 0021/2025/ITP1), CIRP Open Fund of Radiation Protection Laboratories (CIRPFSHJ2025003), Open Fund of National Key Laboratory of Information Systems Engineering (WDZC20255290407), and the High-Performance Computing Center of Central South University.

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

# A    Related Work

## A.1    Traffic Signal Pre-emption for EMVs

To accelerate the clearance of EMVs, early studies [18, 19] adopt immediate green-phase switching upon EMVs detection. This kind of greedy control ensures low delay for EMVs but may heavily disrupt regular traffic. To mitigate this, fuzzy logic-based strategies have been explored. These approaches interpret traffic conditions (e.g., vehicle delays, queues, congestion levels) through fuzzy sets and apply rule-based logic grounded in expert knowledge [20, 21]. Comparative studies suggest fuzzy systems achieve better trade-offs than simpler methods. Optimization-based frameworks have also been proposed to handle signal scheduling more precisely. For instance, elastic signal preemption (ESP) [22] reformulates the problem into a quadratic program triggered periodically as EMVs report their positions. Though potentially more effective, such methods are computationally demanding and may not scale well in practice. In contrast, learning-based methods provide a practical alternative by using trained models to approximate optimal behaviors without the need for frequent online computation, supporting both EMVs efficiency and overall traffic stability.

## A.2    Rl-base Traffic Signal Control

The core objective of adaptive traffic signal control is to dynamically adjust phase sequences to alleviate urban congestion. Early approaches, such as Fixed Time [12] and MaxPressure [13], rely on handcrafted heuristics. Although these strategies are computationally lightweight and interpretable, their dependence on strict assumptions limits their adaptability under complex or dynamic traffic conditions. To address these limitations, recent studies have reformulated traffic signal control as a reinforcement learning (RL) problem, where each intersection acts as an autonomous agent. PPO-TSC [23] designs a PPO-based adaptive controller for single intersections, CoLight [14] introduces graph attention to facilitate inter-agent cooperation, and PressLight [24] embeds max-pressure into the reward function to enhance efficiency. Building on this line of work, several methods further emphasize the importance of modeling temporal and spatiotemporal dependencies to improve coordination stability and responsiveness in dynamic environments [25, 26, 27]. In parallel, trajectory prediction techniques [28, 29] have been integrated into traffic control frameworks to provide fine-grained forecasts of vehicle movements. By anticipating future trajectories, such approaches enable proactive adjustments to signal timing and routing decisions, thereby improving long-term coordination and robustness. Moreover, RL approaches incorporating attention mechanisms [30, 31] and role representations [32, 33, 34] have shown strong potential for capturing inter-agent heterogeneity and learning adaptive communication patterns. These methods enhance cooperation in complex multi-agent environments but remain orthogonal to our focus on the joint coordination of emergency vehicles (EMVs) and traffic lights. Unlike existing studies, our approach explicitly balances the rapid passage of EMVs with the overall efficiency of regular traffic.

# B    Proof of Dynamic Role Learning

To prove that the mutual information objectives in Eq. 5 and Eq. 6 serve as valid training objectives for role representation learning, we follow a formulation inspired by the Mutual Information Neural Estimation (MINE) framework [35], which provides a tractable variational lower bound on mutual information.

Given two random variables $\boldsymbol{h}^{(t)}$ and $\boldsymbol{h}_{emv}^{(t)}$ representing an agent's role embedding and its associated EMV-related features, respectively, their mutual information is defined as:

$$I^{(t)}(\boldsymbol{h}^{(t)}; \boldsymbol{h}_{emv}^{(t)}) = D_{KL}(p(\boldsymbol{h}^{(t)}, \boldsymbol{h}_{emv}^{(t)}) \parallel p(\boldsymbol{h}^{(t)})p(\boldsymbol{h}_{emv}^{(t)}) \tag{16}$$

Using the KL divergence, we obtain the following variational lower bound:

$$I^{(t)}(\boldsymbol{h}^{(t)}; \boldsymbol{h}_{\text{emv}}^{(t)}) \geq \mathbb{E}_{p(\boldsymbol{h}^{(t)}, \boldsymbol{h}_{\text{emv}}^{(t)})}[T_\omega(\boldsymbol{h}^{(t)}, \boldsymbol{h}_{\text{emv}}^{(t)}] - \log \mathbb{E}_{p(\boldsymbol{h}^{(t)})p(\boldsymbol{h}_{\text{emv}}^{(t)})}[\exp(T_\omega(\boldsymbol{h}^{(t)}, \boldsymbol{h}_{\text{emv}}^{(t)}))] \tag{17}$$

Here, $T_\omega$ is a learnable critic function parameterized by $\omega$. The first expectation is taken over positive pairs sampled from the joint distribution, and the second over negative pairs sampled from the product of marginals.

In practice, we estimate the bound with empirical samples $\{(\boldsymbol{h}_j^{(t)}, \boldsymbol{h}_{\text{emv}}^{(t)})\}_{j=1}^{N_n^{(t)}}$, yielding:

$$I^{(t)}(\boldsymbol{h}^{(t)}; \boldsymbol{h}_{\text{emv}}^{(t)}) \geq \frac{1}{N_n^{(t)}} \sum_{i=1}^{N_n^{(t)}} \left[ T_\omega(\boldsymbol{h}_i^{(t)}, \boldsymbol{h}_{\text{emv}}^{(t)}) - \log \left( \frac{1}{N_n^{(t)}} \sum_{j=1}^{N_n^{(t)}} \exp \left( T_\omega(\boldsymbol{h}_j^{(t)}, \boldsymbol{h}_{\text{emv}}^{(t)}) \right) \right) \right]. \tag{18}$$

This empirical form corresponds directly to Eq. 5 in the main text. A similar derivation applies to the trajectory objective in Eq. 6.

## C  Experiment Details

### C.1  Dataset Details

Table 3 presents the main attributes of the datasets used in our experiments, including the total amount of intersections, and the amount of 2-arm to 4-arm intersections. To evaluate the adaptability of our method across diverse road structures, we conduct experiments on the Cologne8 dataset, which contains both standard four-arm intersections and irregular junctions with asymmetric lane configurations. The results confirm the robustness and general applicability of our method in real-world traffic networks.

Table 3: Statistics of the Datesets.

| Datasets | Country | Type | #Total Int. | #2-arm | #3-arm | #4-arm |
|---|---|---|---|---|---|---|
| Grid 4×4 | synthetic | region | 16 | 0 | 0 | 16 |
| Avenue 4×4 | synthetic | region | 16 | 0 | 0 | 16 |
| Cologne8 | Germany | region | 8 | 1 | 3 | 4 |
| Fenglin | China | corridor | 7 | 0 | 2 | 5 |

### C.2  Baseline Details

To benchmark the performance of our approach, we compare it against a variety of methods, including traditional rule-based strategies and modern multi-agent reinforcement learning (MARL) algorithms:

**Traditional Control Strategies.**

- **Fixed-Time (FT)** [12]: Change traffic signals phase in a cyclical sequence with pre-set phase durations and randomly assigned offsets, independent of real-time traffic dynamics.
- **MaxPressure (MP)** [13]: A classical queue-aware approach that greedily selects the phase with the highest pressure, where pressure is defined as the difference between the sum of incoming and outgoing lane queues. It is widely regarded as a strong rule-based traffic control baseline.

**Multi-Agent Reinforcement Learning Approaches.**

- **CoLight** [14]: Utilizes Graph Attention Networks (GAT) to capture spatial dependencies across neighboring intersections. Agents incorporate nearby traffic conditions into their decision-making to minimize local congestion and promote coordinated actions.
- **IPPO** [15]: Assigns an independent PPO agent to each intersection. Although agents share the same network architecture, they are trained using local observations and rewards, allowing decentralized learning.
- **rMAPPO** [16]: Enhances IPPO by integrating global traffic information during training, enabling agents to learn more coordinated strategies across the entire network via centralized value functions.
- **X-Light** [9]: Applies a hierarchical Transformer structure to capture both local intersection dynamics and global decision patterns, enabling effective policy generalization to unseen traffic networks.
- **RECAL** [7]: Applies deep reinforcement learning to dynamically manage traffic signals for emergency vehicles, ensuring rapid passage while mitigating negative impacts on vehicles from conflicting directions.
- **EMVLight** [6]: Employs a decentralized reinforcement learning framework to jointly optimize emergency vehicle routing and traffic signal control, dynamically updating EMV routes in real time while coordinating traffic signals to reduce both EMV travel time and average travel time for non-emergency vehicles.

## D  Model Details

Our complete training process is shown in Algorithm 1.

## E  implementtation Details

We implement all baseline methods using their official codebases and adhere to the recommended settings outlined in their respective papers to ensure fair comparisons. We employ the Adam optimizer with a learning rate of 0.0005. The hidden layer dimension is set to 64. The PPO clip parameter is fixed at 0.2, and the Generalized Advantage Estimation (GAE) lambda parameter is set to 0.95. All experiments are conducted on a single NVIDIA GeForce RTX 3090 GPU with 24GB of memory.

---

**Algorithm 1** Pseudocode for RMTC

---

1: **Input:** $\pi_\theta^{\text{rev}}, \pi_\theta^{\text{tl}}, \pi_\theta^{\text{emv}}$: Policy networks for regular vehicles, lights and emergency vehicles
2: $\quad\quad\quad V_\phi^{\text{rev}}, V_\phi^{\text{tl}}, V_\phi^{\text{emv}}$: Value networks for regular vehicles, lights and emergency vehicles
3: $\quad\quad\quad f_\psi$: Heterogeneous graph-base role encoder
4: $\quad\quad\quad \mathcal{B}$: Replay buffer
5: $\quad\quad\quad T_{cl}$: Time interval for updating Heterogeneous graph convolutional network
6: **for** each episode **do**
7: $\quad$ Reset environment
8: $\quad$ Clear interaction records $\mathcal{B} \leftarrow \emptyset$
9: $\quad$ **for** each timestep $t = 1$ to $T$ **do**
10: $\quad\quad$ Construct heterogeneous graph $G^{(t)} = (\mathcal{I}, E^{(t)})$
11: $\quad\quad$ Compute role embeddings: $\boldsymbol{h}^{(t)} \leftarrow f_\psi(\boldsymbol{o}^{(t)})$
12: $\quad\quad$ **for** agent $i = 1, 2, ..., n$ **do**
13: $\quad\quad\quad$ Sample action $a_i^{(t)} \sim \pi_\theta(\cdot|\boldsymbol{o}_i^{(t)}, \boldsymbol{h}_i^{(t)})$
14: $\quad\quad\quad$ Execute $a_i^{(t)}$ and observe extrinsic reward $r_{i,\text{extr}}^{(t)}$
15: $\quad\quad\quad$ Compute intrinsic reward $r_{i,\text{intr}}^{(t)}$
16: $\quad\quad\quad$ Combine rewards $r_i^{(t)}$
17: $\quad\quad$ **end for**
18: $\quad\quad$ Store the trajectory to in $\mathcal{B}$
19: $\quad$ **end for**
20: $\quad$ **if** episode mod $T_{cl} = 0$ **then**
21: $\quad\quad$ Compute dynamic role loss $\mathcal{L}_{\text{role}}$
22: $\quad\quad$ Update role encoder $f_\psi$ using $\mathcal{L}_{\text{role}}$
23: $\quad$ **end if**
24: $\quad$ Update $\pi_\theta, V_\phi$
25: **end for**

---

# F  Additional Experiments

## F.1  Hyperparameter Analysis

We conducted a hyperparameter analysis of the balance coefficient $\lambda$, which controls the weighting between the intrinsic reward and the environment reward in our framework. As the reward magnitudes for EMVs and regular vehicles are comparable, we fix $\lambda = 1$ for regular vehicles. However, the reward scale between EMVs and traffic signal agents differs significantly. Therefore, we analyze the performance impact of varying $\lambda$ for signal agents in the range of $\{0.001, 0.005, 0.01, 0.05, 0.1\}$.

As shown in Figure 5, the overall performance first improves with increasing $\lambda$, reaches a peak at $\lambda = 0.01$, and then begins to decline. This indicates that a moderate emphasis on the intrinsic objective is beneficial, while over-weighting it may lead to performance degradation. Based on this analysis, we set $\lambda = 0.01$ for traffic signal agents in all experiments.

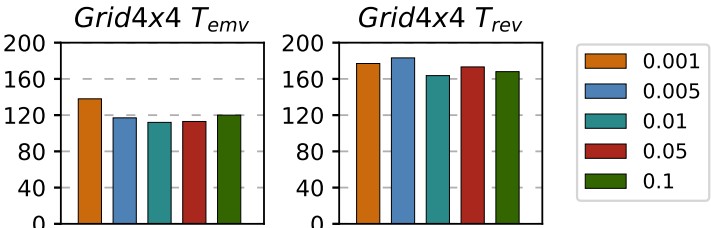

Figure 5: Parameter sensitivity analysis of the balance coefficient $\lambda$ for signal agents on EMVs travel time and REVs travel time.

Table 4: Ablation study on RMTC showing episode time, inference time, and reward for different signal settings.

| Signal | Method | **Episode Time (s)** | **Inference Time (s)** | Reward |
|---|---|---|---|---|
| signal-8 | RMTC | 122.12 | 0.0275 | -0.561 |
| | w/o GNN | 101.25 | 0.0269 | -0.780 |
| | w/o MI | 106.09 | 0.0235 | -0.665 |
| signal-16 | RMTC | 160.90 | 0.0424 | -0.144 |
| | w/o GNN | 130.97 | 0.0396 | -0.157 |
| | w/o MI | 142.02 | 0.0422 | -0.275 |
| signal-25 | RMTC | 239.92 | 0.0507 | -0.487 |
| | w/o GNN | 222.61 | 0.0421 | -0.511 |
| | w/o MI | 236.00 | 0.0469 | -1.229 |

## F.2 Case Study

To further evaluate the role mechanism and its influence on agent behavior, we conduct a case study using the Cologne8 dataset. Specifically, we visualize the trajectories of an EMV from its entry into the road network to its arrival at the destination at two time steps. As shown in Figure 6, the EMV's trajectory is highlighted in red. Vehicles highlighted in dark blue represent those influenced by the EMV, while light blue denotes all other vehicles.

The visualized trajectories reveal that vehicles in close proximity to the EMV in space or along its path tend to steer clear of the EMV's trajectory to prevent potential conflicts. In contrast, vehicles farther away do not exhibit yielding behavior, indicating no obvious response to the EMV. This distinction illustrates that the learned role embeddings enable local agents to perceive and respond to EMV presence, while preserving normal behavior in unaffected areas.

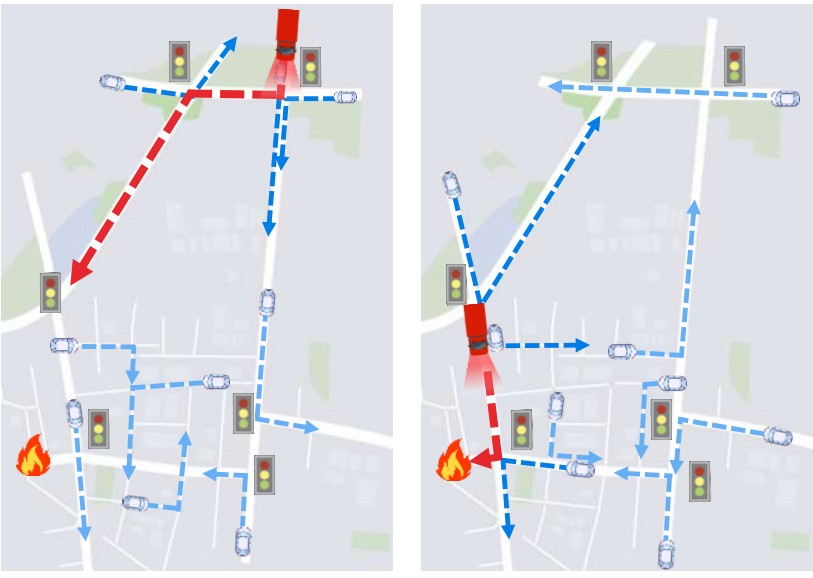

Figure 6: Visualization of EMV and REV Trajectories from EMV Entry to Destination.

## F.3 Efficiency Analysis

We report episode training time and inference time averaged over 100 episodes for traffic networks with 8, 16, and 25 traffic signals. As shown in the table below, the episode time of RMTC increases from 122.12s (8 signals) to 160.90s (16 signals) and 239.92s (25 signals). The inference time similarly grows from 0.0275s to 0.0424s and 0.0507s. Despite the increasing network size, the growth trend remains close to linear, demonstrating that RMTC can scale efficiently with larger traffic scenarios. Removing the GNN module reduces training time by 35.7%

(16-signal case), and the MI component increases training time by 13%, with minimal impact on inference. Despite the added cost, our full model achieves the highest final reward among all variants, demonstrating better learning performance without hindering practical deployment.

# G   Limitation

One practical limitation of our approach is that it relies on the presence of emergency events. The proposed coordination strategy is specifically designed to respond to EMVs; it is only when such events occur that the role-aware coordination mechanism is activated and contributes to improved traffic flow. Another limitation is the assumption that all vehicles strictly follow their assigned navigation routes. In practice, drivers may deviate from recommended paths due to personal preferences or external constraints, potentially reducing the effectiveness of our coordinated control strategy.

