# OpenReview forum: "Role-aware Multi-agent Reinforcement Learning for Coordinated Emergency Traffic Control"
_NeurIPS.cc/2025/Conference — NeurIPS 2025 poster_

### Official Review · Reviewer_2Wmh · 2025-06-25

**Clarity:** 3
**Significance:** 4
**Originality:** 4
**Rating:** 5
**Confidence:** 4

**Summary:**

This paper proposes Role-aware Multi-agent Traffic Control (RMTC), a novel framework designed to enhance traffic coordination in scenarios involving emergency vehicles. RMTC introduces a Heterogeneous Temporal Traffic Graph (HTTG) to comprehensively capture the spatial-temporal interactions among traffic lights, regular vehicles, and emergency vehicles. Building on this representation, the framework incorporates a Dynamic Role Learning module that adaptively infers the roles of different traffic agents based on evolving traffic conditions. These roles are then leveraged in a Role-aware Multi-agent Reinforcement Learning algorithm to guide decision-making and coordination. Through extensive evaluations on four benchmark scenarios, RMTC demonstrates superior performance in reducing emergency vehicle delays while maintaining overall traffic efficiency, highlighting the benefits of incorporating dynamic roles into multi-agent traffic control.

**Questions:**

Q1. Could the authors explain how role features are used to produce distinct behaviors across roles, and how this is reflected in the policy learning process?

Q2. Please provide more detailed descriptions for the training of the Dynamic Role Learning module. This would help readers better understand and reproduce the approach.

Q3. How does the framework handle scenarios with multiple emergency vehicles? Specifically, how is the intrinsic reward in Eq.11 extended or adapted to such cases? Clarification on this point would strengthen the technical rigor of the paper.

**Ethical Concerns:**

["NO or VERY MINOR ethics concerns only"]

**Limitations:**

yes

**Paper Formatting Concerns:**

No major formatting problems were found.

**Quality:**

4

**Strengths And Weaknesses:**

Strengths:
S1. The paper is generally well-organized, with a logical flow of methodology. The equations and definitions are formal and clear.

S2. The authors explicitly formulate the coordination of emergency vehicles, regular vehicles, and traffic lights as a joint multi-agent decision-making problem, which is a relatively new perspective in traffic control and provides new insights for related research.

S3. The proposed integration of a heterogeneous temporal traffic graph and role-aware reinforcement learning is reasonable and effective. The experiments are sufficient, the settings and data preparation are well-illustrated, and the results are well-discussed.

Weaknesses:
W1. The paper introduces role-aware reinforcement learning, but it is unclear how different roles actually lead to different behavior patterns in agents.

W2. The multi-agent modeling does not provide enough details, especially about how the Dynamic Role Learning module is trained, making it hard to fully understand or reproduce.

W3. Some formulas in the manuscript are incomplete or simplified. For example, Eq.11 assumes only one EMV in the scenario, but the paper does not clarify how the intrinsic reward would be computed when multiple EMVs are present.

---

> ### Author Rebuttal · Authors · 2025-07-31
>
> First and foremost, we sincerely thank you for pointing out the issues. Your suggestions are invaluable in enhancing the quality of this paper. Below is our answer to your questions.
>
> > W1&Q1.  The paper introduces role-aware reinforcement learning, but it is unclear how different roles actually lead to different behavior patterns in agents.
>
> Thank you for your comments. We address them from two perspectives:
>
> - **Role Influence on Policy:** Dynamic roles affect RL behavior in two ways: role embeddings are combined with local states as inputs to the policy network, enabling EMV-aware decisions. Additionally, a role-conditioned intrinsic reward encourages agents to take EMV-friendly actions. Together, these mechanisms guide both behavior and coordination, improving training efficiency.
> - **Case Study:** Following the reviewer’s comment, we showcase how the dynamic role influences the reinforcement learning process with the following case. TL 1, one hop ahead of the EMV, has a high cosine similarity (0.49) to the EMV feature and maintains its green phase to facilitate passage. TL 2, with no nearby EMV, shows low similarity (0.17) and changes its signal independently. This shows role embeddings capture EMV proximity and influence signal behavior accordingly.
>
> | Agent | Relative to EMV | Cosine Sim (Role vs EMV feat) | Action                                 |
> | ----- | --------------- | ----------------------------- | -------------------------------------- |
> | TL 1  | EMV 1 hop ahead | 0.49                          | gGGGgrrrrgGGGgrrrr->gGGGgrrrrgGGGgrrrr |
> | TL 2  | No EMV nearby   | 0.17                          | gGGGgrrrrgGGGgrrrr->grrrgGGGGgrrrgrrrr |
>
> *Action indicates the signal phase before and after the decision. Each character corresponds to a lane direction: 'g' denotes green and 'r' denotes red. For TL 1, the EMV enters when the signal is green, and the phase remains unchanged to allow passage.*
>
> > W2&Q2. The multi-agent modeling does not provide enough details, especially about how the Dynamic Role Learning module is trained, making it hard to fully understand or reproduce.
>
> Thank you for your comments.
>
> - **Role Embedding Learning:** We first train the Heterogeneous Role Convolution module using the role learning objective in Eq.9. This module captures spatial-temporal interactions in the heterogeneous traffic graph and outputs dynamic role embeddings for each agent. These embeddings are learned to reflect agent role, especially in relation to nearby EMVs.
>
> - **Role-aware Policy Learning:** The learned role embeddings are then concatenated with the agent’s state to form role-aware observations. These enhanced inputs are fed into the policy networks, allowing each agent to condition its decisions on both local traffic state and semantic role information. Role learning and policy optimization proceed in an alternating training loop to ensure consistency and coordination.
>
> > W3&Q3. Some formulas in the manuscript are incomplete or simplified. For example, Eq.11 assumes only one EMV in the scenario, but the paper does not clarify how the intrinsic reward would be computed when multiple EMVs are present.
>
> Thank you for your valuable suggestion. Our method encodes the relations between each agent and multiple EMVs through a heterogeneous graph, learning role embeddings via mutual information maximization. This design ensures that agent roles dynamically reflect their varying relevance to different EMVs based on spatial positions. Additionally, we propose a role-similarity-based reward redistribution mechanism to enable soft prioritization when multiple EMVs coexist. We have revised Eq.11 to the following form to support multi-EMV scenarios:
>
> $r\_{n,\text{intr}}^{(t)} = \sum\_{i=1}^{N_{emv}} s\left(h\_n^{(t)}, h\_{\text{emv},m}^{(t)}\right) \cdot r\_{\text{emv},i,\text{extr}}^{(t)}, \quad \forall n \in \{ I_{\text{rev}}, I_{\text{tl}} \}$
>
> We will update Eq.11 and the corresponding explanations in the manuscript to clearly present this multi-EMV reward computation.

---

> > ### Comment · Reviewer_2Wmh · 2025-08-02
> >
> > Thank you very much for your thoughtful response and the additional analysis. Most of my concerns have been resolved.
> > I appreciate the idea and motivation behind this work. It addresses a novel and practically important problem in multi-agent traffic coordination. Hence, I would like to keep my initial score.

---

> > > ### Author Response · Authors · 2025-08-02
> > > **Thank you for your reply**
> > >
> > > Thank you for your thoughtful review and feedback. We're glad our response addressed most of your concerns, and we appreciate your continued support for our paper.

---

### Official Review · Reviewer_5jVo · 2025-06-30

**Clarity:** 2
**Significance:** 3
**Originality:** 3
**Rating:** 4
**Confidence:** 4

**Summary:**

The paper introduces RMTC, a role-aware multi-agent reinforcement learning framework for emergency traffic control. It dynamically assigns roles to traffic lights and vehicles using a Temporal Traffic Graph, enabling coordinated behavior based on EMV position and trajectory. The method is tested against several benchmarks demonstrating an improvement in the EMV travel time without hurting the overall traffic efficiency.

**Questions:**

1. Mainly about the sensitivity and scalability of the method, I am not sure how well it would scale to a real-life problem and if not what are the limitations.

**Ethical Concerns:**

["NO or VERY MINOR ethics concerns only"]

**Final Justification:**

This is a usufull paper. I retain my score after rebuttal.

**Limitations:**

yes

**Paper Formatting Concerns:**

The paper is well-structured and clear. One aspect that can be improved is the description of the reward signals.

**Quality:**

2

**Strengths And Weaknesses:**

Strengths:
1. Interesting framework, unifying both signal control and navigation.
2. Results support the claim that this approach can benefit EMVs and should be explored further.

Weaknesses:
1. Limited modeling as EMVs are part of the system and treated as such, instead of allowing for individual decisions.
2. Scalability of the method is unclear. two aspects are relevant, how the method scales with the number of intersections (that seems to scale at least to small-medium networks), and how it scales with growing number of EMVs.

---

> ### Author Rebuttal · Authors · 2025-07-31
>
> First and foremost, we sincerely thank you for pointing out the issues. Your suggestions are invaluable in enhancing the quality of this paper. Below is our answer to your questions.
>
> > W1. Limited modeling as EMVs are part of the system and treated as such, instead of allowing for individual decisions.
>
> Thank you for the comment. In our framework, EMVs are modeled as independent learning agents with their own observations, policies, and rewards. This allows each EMV to make individual decisions rather than being passively controlled by the environment or rule-based logic.
>
> > W2 & Q1. Scalability and Real-World Applicability
>
> Thank you for your comment. We address this from the following four perspectives:
>
> - **New experiment on Larger traffic Network Size.** Following prior works[1,2,3], we conduct our main experiments on commonly used traffic networks. To further assess scalability with respect to the number of intersections, we introduce a larger 5×5 grid network comprising 25 intersections. The results show that our method consistently maintains effective performance and coordination as the network size increases.
> - **Computational Overhead**. We report episode training time and inference time averaged over 100 episodes for traffic networks with 8, 16, and 25 traffic signals. As shown in the table below, the episode time of RMTC increases from 122.12s (8 signals) to 160.90s (16 signals) and 239.92s (25 signals). The inference time similarly grows from 0.0275s to 0.0424s and 0.0507s. Despite the increasing network size, the growth trend remains close to linear, demonstrating that RMTC can scale efficiently with larger traffic scenarios.
> - **Impact of GNN and MI on Computation:** Removing the GNN module reduces training time by 35.7% (16-signal case), and the MI component increases training time by 13%, with minimal impact on inference. Despite the added cost, our full model achieves the highest final reward among all variants, demonstrating better learning performance without hindering practical deployment.
> - **Scalability to Large Real-World Scenarios**. In practice, EMV dispatch is organized by region, as emergency services such as fire stations and hospitals are distributed across districts. Our method supports region-wise training and deployment, making it well-suited for scalable application in large cities, rather than requiring training over the entire city.
>
> **Comparison on Grid 5×5**
>
> | Method | Grid 5×5 $T_{emv}$ | Grid 5×5 $T_{ve}$ |
> | ------ | ------------------ | ----------------- |
> | FT     | 551                | 628.49            |
> | mappo  | 223                | 427.24            |
> | RMTC   | 199                | 410.91            |
>
> **Efficiency experiment**
>
> |           |         | **Episode Time** (s) | **Inference Time**(s) | Reward |
> | --------- | ------- | -------------------- | ---------------------- | ------ |
> | signal-8  | RMTC    | 122.12               | 0.0275                 | -0.561 |
> |           | w/o GNN | 101.25               | 0.0269                 | -0.780 |
> |           | w/o MI  | 106.09               | 0.0235                 | -0.665 |
> | signal-16 | RMTC    | 160.90               | 0.0424                 | -0.144 |
> |           | w/o GNN | 130.97               | 0.0396                 | -0.157 |
> |           | w/o MI  | 142.02               | 0.0422                 | -0.275 |
> | signal-25 | RMTC    | 239.92               | 0.0507                 | -0.487 |
> |           | w/o GNN | 222.61               | 0.0421                 | -0.511 |
> |           | w/o MI  | 236.00               | 0.0469                 | -1.229 |
>
> [1]Jiang, Haoyuan, et al. "X-Light: cross-city traffic signal control using transformer on transformer as meta multi-agent reinforcement learner." *Proceedings of the Thirty-Third International Joint Conference on Artificial Intelligence*. 2024.
>
> [2]Wei, Hua, et al. "Colight: Learning network-level cooperation for traffic signal control." *Proceedings of the 28th ACM international conference on information and knowledge management*. 2019.
>
> [3]Wei, Hua, et al. "Presslight: Learning max pressure control to coordinate traffic signals in arterial network." *Proceedings of the 25th ACM SIGKDD international conference on knowledge discovery & data mining*. 2019.

---

> > ### Comment · Reviewer_5jVo · 2025-08-06
> >
> > Thank you for your detailed response and the effort in addressing my comments. While some aspects have been clarified, my overall evaluation remains unchanged. I will therefore maintain my original score.

---

### Official Review · Reviewer_HR9y · 2025-07-02

**Clarity:** 3
**Significance:** 3
**Originality:** 3
**Rating:** 3
**Confidence:** 4

**Summary:**

This paper proposes a novel framework called RMTC (Role-aware Multi-agent Traffic Control) that coordinates emergency vehicles (EMVs), regular vehicles (REVs), and traffic lights for efficient traffic control during emergencies. It introduces a Heterogeneous Temporal Traffic Graph (HTTG) to represent spatial-temporal relations among traffic agents and employs a dynamic role-learning mechanism to adaptively assign behaviors. A role-aware reinforcement learning model is trained using both extrinsic and intrinsic rewards derived from EMV influence. Experiments on four datasets (synthetic and real-world) show improved EMV travel time and overall traffic efficiency compared to baselines.

**Questions:**

1) While the proposed role-learning loss appears intuitively plausible, it remains largely heuristic and lacks theoretical justification. In particular, the paper does not explain why maximizing mutual information between role embeddings and EMV state features would necessarily lead to better coordination or policy learning.  Can you showcase how the dynamic role influences the reinforcement learning process? 2) The paper introduces a role embedding mechanism, but lacks evidence that these embeddings are meaningful or useful. Could you please elaborate further on this point? 3)  The interaction and potential conflicts between the different loss components (e.g., role consistency vs. responsiveness to EMVs) are not analyzed or theoretically grounded. 4) While the proposed framework involves reinforcement learning as a core optimization engine, the paper lacks critical evidence of actual policy learning. There are no training curves, no sample efficiency comparisons, and no analyses of training stability, convergence, or the effects of reward shaping.

**Ethical Concerns:**

["NO or VERY MINOR ethics concerns only"]

**Final Justification:**

I appreciate the authors’ efforts in preparing the thoughtful rebuttal. I have carefully reviewed the authors' rebuttal and the comments of other reviewers. The responses have addressed some of the questions, but do not fully address my concerns: 1) the effectiveness and efficiency of the RL engine, 2) the impacts of the learned embedding. These unsolved concerns are central to the effectiveness and practical significance of this work. These unresolved concerns weigh heavily in my overall assessment.  Thus, my overall recommendation for this work remains unchanged.

**Limitations:**

Yes, the authors discussed limitations in the appendix. However, it would be better to discuss some limitations in the main body.

**Paper Formatting Concerns:**

No major formatting issue.

**Quality:**

3

**Strengths And Weaknesses:**

The studied problem is interesting and important. The strengths: 1) Problem Setup: Formulates a novel multi-agent emergency traffic control problem that jointly coordinates EMVs, REVs, and traffic lights, beyond conventional traffic light-only control; 2) HTTG + Role Learning: Proposes the Heterogeneous Temporal Traffic Graph to model time-varying interactions and designs a dynamic role learning module based on mutual information maximization and role consistency; 3) Role-aware Reinforcement Learning: Combines role-conditioned observations with intrinsic and extrinsic rewards to guide traffic control policies: 4) Empirical Validation: Demonstrates clear improvements on EMV and REV travel time across multiple datasets, including ablation and stress testing under traffic load variations. Weaknesses: 1) The related work section is not comprehensive and overlooks many recent studies. 2) It’s not clear either in Figure 2 or the main content how the dynamic role influences the reinforcement learning process; 3) There is no latest traffic signal control baseline methods or methods specifically designed for traffic signal control with emergency vehicles, such as EMVLight or RECAL; 4) The paper does not report variance or error bars across different random seeds, limiting confidence in the reliability of performance gains; 5) The number of agents used in each task is not specified. If each vehicle requires an independent policy, the computational burden could be substantial. This important detail should be clearly addressed in the paper.

---

> ### Author Rebuttal · Authors · 2025-07-31
>
> First and foremost, we sincerely thank you for pointing out the issues. Your suggestions are invaluable in enhancing the quality of this paper. Below is our answer to your questions.
>
> > W1. The related work section is not comprehensive and overlooks many recent studies.
>
> Thank you for your helpful comment. In the revised version, we will expand the related work section to include recent advances in traffic signal control and studies specifically addressing emergency vehicle scenarios[1,2,3], which are closely related to our problem setting.
>
> > W2 & Q2 & Q1. How are dynamic role embeddings learned and how do they influence the reinforcement learning process?
>
> Thank you for your comments. We address them from three perspectives:
>
> - **Role Embedding Learning:** Our approach learns compact role embeddings that focus on essential EMV-related information by maximizing the mutual information (MI) between the embeddings and the local EMV state. This ensures the roles capture task-critical semantic cues, serving as informative context variables that help the policy network concentrate on critical features. Prior works such as ROMA [4], RODE [5], and ACORM[6] demonstrate that maximizing mutual information between role embeddings and key state features improves policy learning and efficiency. This supports our use of EMV-centered mutual information to enhance role semantics and policy performance.
>
> - **Role Influence on Policy:** Dynamic roles affect RL behavior in two ways: role embeddings are combined with local states as inputs to the policy network, enabling EMV-aware decisions. Additionally, a role-conditioned intrinsic reward encourages agents to take EMV-friendly actions. Together, these mechanisms guide both behavior and coordination, improving training efficiency.
>
> - **Case Study:** Following the reviewer’s comment, we showcase how the dynamic role influences the reinforcement learning process with the following case. TL 1, one hop ahead of the EMV, has a high cosine similarity (0.49) between its role embedding and the EMV feature, which reflects the level of EMV-awareness and contextual relevance. TL 2, with no nearby EMV, shows low similarity (0.17) and changes its signal independently. This shows role embeddings capture EMV proximity and influence signal behavior accordingly.
>
>   | Agent | Relative to EMV | Cosine Sim (Role vs EMV feat) | Action                                 |
>   | ----- | --------------- | ----------------------------- | -------------------------------------- |
>   | TL 1  | EMV 1 hop ahead | 0.49                          | gGGGgrrrrgGGGgrrrr->gGGGgrrrrgGGGgrrrr |
>   | TL 2  | No EMV nearby   | 0.17                          | gGGGgrrrrgGGGgrrrr->grrrgGGGGgrrrgrrrr |
>
> *Action indicates the signal phase before and after the decision. Each character corresponds to a lane direction: 'g' denotes green and 'r' denotes red. For TL 1, the EMV enters when the signal is green, and the phase remains unchanged to allow passage.*
>
> > W3&W4. Baselines and performance stability
>
> Thanks for your suggestion. To address recent concerns, we include EMV-specific baselines, and report mean ± standard deviation over 3 random seeds for our method.
>
>
> | Method   | grid4*4 $T_{emv}$ | grid4*4 $T_{ve}$ | avenue $T_{emv}$ | avenue $T_{ve}$ | cologne $T_{emv}$ | cologne $T_{ve}$ | Fenglin $T_{emv}$ | Fenglin $T_{ve}$ |
> | -------- | -------------- | ------------- | ---------------- | --------------- | ----------------- | ---------------- | ----------------- | ---------------- |
> | RECAL    | 129            | 296.39        | 162              | 757.83          | 47                | 159.93           | 38                | 276.91           |
> | EMVlight | 117            | 176.03        | 254              | 625.85          | 47                | 148.9            | 34                | 293.79           |
> | RMTC     | 108±9.64       | 162.71±0.87   | 151.33±11.67     | 508.47±18.87    | 46±8.08           | 89.02±4.84       | 30.66±3.29        | 225.61±5.72      |
>
> > W5. The number of agents used in each task is not specified. If each vehicle requires an independent policy, the computational burden could be substantial. This important detail should be clearly addressed in the paper
>
> Thank you for pointing this out.
>
> - **Vehicle agent number setting.** In our framework,  each vehicle and traffic light is modeled as an agent. The number of vehicles is determined by the vehicle generation rate and episode duration. Specifically, the Grid and Cologne datasets contain approximately 600 vehicles, while the Arterial and Fenglin datasets include around 1000 vehicles per episode.
>
> - **Shared policy.** All agents share a centralized policy network (parameter sharing). This design enables agent-specific behavior while avoiding the computational burden of maintaining independent policies for each agent, ensuring scalability and learning efficiency in large-scale environments. We will include a detailed explanation of this design choice in the final version
>
> > Q3. The interaction and potential conflicts between the different loss components (e.g., role consistency vs. responsiveness to EMVs) are not analyzed or theoretically grounded
>
> Thank you for the suggestion. We address the concern from two aspects:
>
> - **Temporal stability**: The temporal consistency loss is designed to ensure smooth role transitions across adjacent time steps. It does not constrain adaptation to major changes. When an EMV changes its road segment, the mutual information loss actively guides the role embedding to adapt, ensuring responsiveness. Thus, the two objectives are functionally complementary rather than conflicting.
>
> - **Experiment**: We experimented with different values of the weight $\alpha$ in the role loss function, which is defined as follows.
>
>   $\mathcal{L}\_{\text{role}} = \sum\_{t=1}^{T}\left( - \alpha \left( I^{(t)}(h^{(t)};h^{(t)}\_{\text{emv}}) + I(h^{(t)}; g\_{\text{emv}}^{(t)}) \right) + \mathcal{L}\_{\text{cons}}^{(t)}\right)$​
>
>   The results show that setting \($\alpha$ = 1\) yields the best balance, achieving the lowest EMV travel time and vehicle travel time. This indicates a stable and effective interaction between the mutual information terms and the consistency loss.
>
> | $\alpha $ |grid4*4 $T_{emv}$ |grid4*4 $T_{ve}$ |
> | --------- | --------- | -------- |
> | 0.5       | 139       | 194.38   |
> | 2         | 133       | 196.15   |
> | 1         | 112       | 163.62   |
>
> > Q4. While the proposed framework involves reinforcement learning as a core optimization engine, the paper lacks critical evidence of actual policy learning. There are no training curves, no sample efficiency comparisons, and no analyses of training stability, convergence, or the effects of reward shaping.
>
> Thank you for the valuable comment. Below, we provide a detailed analysis of RMTC’s training stability, convergence, and sample efficiency:
>
> - The first table shows the episodic rewards of RMTC and its ablations across different training stages. RMTC consistently achieves higher rewards and faster convergence. Stability is measured by the standard deviation of evaluation rewards, where RMTC exhibits the lowest standard deviation (1.480), indicating superior stability.
> - The second table compares RMTC and MAPPO across different timesteps. RMTC achieves better performance at each stage, especially in early training (e.g., at 5k and 10k timesteps), demonstrating stronger sample efficiency and faster learning.
>
> **Reward Ablation Study on RMTC Components Across Episodes**
>
> |                                   | Episode10 | Episode20 | Episode50 | Episode100 | Stability |
> | --------------------------------- | --------- | --------- | --------- | ---------- | --------- |
> | RMTC                              | -0.509    | -0.222    | -0.139    | -0.119     | 1.480     |
> | w/o EMV Position Role Impacting   | -1.393    | -0.770    | -0.301    | -0.289     | 2.387     |
> | w/o Role Consistency Constraint   | -1.237    | -0.644    | -0.511    | -0.203     | 3.205     |
> | w/o EMV Trajectory Role Impacting | -1.152    | -0.293    | -0.185    | -0.187     | 3.147     |
> | w/o Role-aware Intrinsic Reward   | -1.198    | -0.407    | -0.232    | -0.278     | 3.099     |
>
> **Sample efficiency Comparison Between RMTC and MAPPO**
>
> |       | Environment steps 5k | Environment steps  10k | Environment steps  25k | Environment steps  50k |
> | ----- | -------------------- | ---------------------- | ---------------------- | ---------------------- |
> | mappo | -1.739               | -0.880                 | -0.1557                | -0.149                 |
> | RMTC  | -0.891               | -0.600                 | -0.1445                | -0.127                 |
>
> [1]Yao, Jiarong, et al. "Incorporating vision-based artificial intelligence and large language model for smart traffic light control." *Applied Soft Computing* (2025): 113333.
>
> [2]Wang, Lijuan, et al. "An adaptive traffic signal control scheme with Proximal Policy Optimization based on deep reinforcement learning for a single intersection." *Engineering Applications of Artificial Intelligence* 149 (2025): 110440.
>
> [3]Alruwaili, Madallah, et al. "LSTM and ResNet18 for optimized ambulance routing and traffic signal control in emergency situations." *Scientific Reports* 15.1 (2025): 6011.
>
> [4]Wang, Tonghan, et al. "ROMA: multi-agent reinforcement learning with emergent roles." *Proceedings of the 37th International Conference on Machine Learning*. 2020.
>
> [5]Wang, Tonghan, et al. "RODE: Learning Roles to Decompose Multi-Agent Tasks." *International Conference on Learning Representations*.
>
> [6]Hu, Zican, et al. "Attention-Guided Contrastive Role Representations for Multi-agent Reinforcement Learning." *The Twelfth International Conference on Learning Representations*.

---

> > ### Comment · Reviewer_HR9y · 2025-08-05
> >
> > Thank you for the detailed response. I appreciate your efforts in preparing the rebuttal to address the comments I mentioned previously. Although the responses have clarified a few aspects of the original submission, they have not changed my overall assessment of this work. Thus, I would like to maintain my score for this work.

---

### Official Review · Reviewer_Q23t · 2025-07-03

**Clarity:** 3
**Significance:** 2
**Originality:** 2
**Rating:** 3
**Confidence:** 3

**Summary:**

The paper proposes RMTC, a role-aware multi-agent RL framework that jointly controls traffic-lights, regular vehicles, and emergency vehicles (EMVs). It builds a heterogeneous temporal traffic graph, learns dynamic role embeddings driven by EMV position/trajectory, and adds a role-aware intrinsic reward. Experiments on two synthetic and two real SUMO scenarios show 4 – 26 % EMV travel-time reduction while keeping regular traffic efficient.

**Questions:**

- Shouldn’t there be a rule-based baseline that prioritizes EMV completely to get an idea about the shortest time for EMV?
- How does RMTC scale up to a large-city map? How to apply this method to real-world scenarios?
- How are the baselines compared? What factors are controlled, and how were hyperparameters tuned separately for each baseline? Please clarify your controlled experiments.
- How to handle contentions among many simultaneous emergencies?

**Ethical Concerns:**

["NO or VERY MINOR ethics concerns only"]

**Limitations:**

- Framework only activates when EMV is detected; benefits vanish in normal conditions.
- Relies on idealized communication, accurate, low-latency global state; noisy detection or non-cooperative drivers may break assumptions.
- It is hard to gauge its contribution to the RL or role-based MARL without experiments in other domains, given that this paper is evaluated in the RL track.

**Quality:**

2

**Strengths And Weaknesses:**

**Strengths**

- Addresses a practically important but under-studied three-party coordination problem.
- Solid technical integration: graph message passing + mutual-information role learning + intrinsic reward.
- Thorough and towards real-world experiments setups: four benchmarks, ablations, congestion, and multi-EMV stress tests, code released.
- Empirical gains over strong RL baselines (rMAPPO, X-Light) on both EMV and regular travel time.

**Weaknesses**

- Novelty incremental: many elements (GNN, MI role learning, PPO) exist in prior MARL/traffic-RL work like [1] and its prior works; contribution is mainly a combination.
- Unclear evaluation:
   - It is unclear how the dataset is split into train & test to report the evaluation metrics.
   - It is unclear how the method performs under normal scenarios
- Scalability unclear beyond ~16 intersections; compute overhead of graph & MI terms not reported.
- Assumes perfect sensing, V2X comms, and strict driver compliance, which may limit its real-world application.
- Related Work:
  - Related work lacks recent attention-based or graph-meta RL traffic control methods and possibly role-based MARL
- Poor writing quality:
  - several typos (“Tough”, “implementtation”)
  - Equation (14), it is unclear whether V is value function of extrinsic reward only or extr+intr
  - missing reference to PPO by shulman et al.,
  - A.2 what is Rl-base → RL-based?

[1] Goel, H., Omama, M., Chalaki, B., Tadiparthi, V., Pari, E. M., & Chinchali, S. (2025). R3DM: Enabling Role Discovery and Diversity Through Dynamics Models in Multi-agent Reinforcement Learning. arXiv preprint arXiv:2505.24265.

---

> ### Author Rebuttal · Authors · 2025-07-31
>
> First and foremost, we sincerely thank you for pointing out the issues. Your suggestions are invaluable in enhancing the quality of this paper. Below is our answer to your questions.
>
> > W1. Incremental contribution and Novelty.
>
> Thank you for your comment. We will clarify our key contributions in the revised version.
>
> - **New Problem Setting**: We formulate emergency traffic coordination as a multi-agent problem involving EMVs, REVs, and traffic lights—going beyond prior TSC works that only control signals.
> - **Framework Design**: Instead of combining existing techniques, we customize MI-based role learning, heterogeneous temporal graphs, and role-conditioned RL to address behavior differentiation and cross-agent interaction in emergency scenarios.
> - **EMV-aware Role Learning**: We further design three losses based on EMV states and trajectories to enable dynamic role representation, and introduce a role-aware intrinsic reward to balance global traffic efficiency with EMV priority.
>
> > W2. Unclear evaluation:
> >
> > Dataset Splitting for Training and Testing
> >
> > Method Applicability in Normal Traffic Scenarios
>
> Thank you for your comment.
>
> - **Evaluation Protocol**: Our evaluation is conducted in the SUMO simulation environment. Unlike supervised learning, reinforcement learning does not rely on static training/testing splits. We evaluates performance by varying initial environment conditions. This approach is widely adopted in traffic control tasks[1,2].
> - **Degraded Setting**: In scenarios without emergency vehicles, our method naturally reduces to a basic actor-critic model without role embeddings or intrinsic rewards.
>
> > W3 & Q2. Scalability, computational overhead, and applicability to large-scale real-world scenarios
>
> Thank you for your comment. We address this from the following four perspectives:
>
> - **New experiment on Larger traffic Network Size.** Following prior works[1,2,3], we conduct our main experiments on commonly used traffic networks. To further assess scalability with respect to the number of intersections, we introduce a larger 5×5 grid network comprising 25 intersections. The results show that our method consistently maintains effective performance and coordination as the network size increases.
> - **Computational Overhead**. We report episode training time and inference time averaged over 100 episodes for traffic networks with 8, 16, and 25 traffic signals. As shown in the table below, the episode time of RMTC increases from 122.12s (8 signals) to 160.90s (16 signals) and 239.92s (25 signals). The inference time similarly grows from 0.0275s to 0.0424s and 0.0507s. Despite the increasing network size, the growth trend remains close to linear, demonstrating that RMTC can scale efficiently with larger traffic scenarios.
> - **Impact of GNN and MI on Computation:** Removing the GNN module reduces training time by 35.7% (16-signal case), and the MI component increases training time by 13%, with minimal impact on inference. Despite the added cost, our full model achieves the highest final reward among all variants, demonstrating better learning performance without hindering practical deployment.
> - **Scalability to Large Real-World Scenarios**. In practice, EMV dispatch is organized by region, as emergency services such as fire stations and hospitals are distributed across districts. Our method supports region-wise training and deployment, making it well-suited for scalable application in large cities, rather than requiring training over the entire city.
>
> **Comparison on Grid 5×5**
>
> | Method | Grid 5×5 $T_{emv}$ | Grid 5×5 $T_{ve}$ |
> | ------ | ------------------ | ----------------- |
> | FT     | 551                | 628.49            |
> | mappo  | 223                | 427.24            |
> | RMTC   | 199                | 410.91            |
>
> **Efficiency experiment**
>
> |           |         | **Episode Time** (s) | **Inference Time**(s) | Reward |
> | --------- | ------- | -------------------- | ---------------------- | ------ |
> | signal-8  | RMTC    | 122.12               | 0.0275                 | -0.561 |
> |           | w/o GNN | 101.25               | 0.0269                 | -0.780 |
> |           | w/o MI  | 106.09               | 0.0235                 | -0.665 |
> | signal-16 | RMTC    | 160.90               | 0.0424                 | -0.144 |
> |           | w/o GNN | 130.97               | 0.0396                 | -0.157 |
> |           | w/o MI  | 142.02               | 0.0422                 | -0.275 |
> | signal-25 | RMTC    | 239.92               | 0.0507                 | -0.487 |
> |           | w/o GNN | 222.61               | 0.0421                 | -0.511 |
> |           | w/o MI  | 236.00               | 0.0469                 | -1.229 |
>
>
> > W4. Assumes perfect sensing, V2X comms, and strict driver compliance, which may limit its real-world application.
>
> We appreciate the reviewer’s concern. Our framework assumes V2X-based communication, where vehicles can exchange information with traffic lights and nearby vehicles. This assumption is widely adopted in EMV coordination research. Additionally, we consider each vehicle’s destination, which allows integration with real-world navigation systems to assist both emergency and regular drivers in cooperative routing during emergencies.
>
> > W5. Related Work: Related work lacks recent attention-based or graph-meta RL traffic control methods and possibly role-based MARL
>
> Thank you for your comment. We will add a discussion of recent works on attention-based or graph-meta reinforcement learning for traffic control [4,5] and role-based multi-agent reinforcement learning methods [6,7,8] to the related work section in the revised version. These methods differ from ours in both problem formulation and data modalities, and thus were not directly compared in our experiments.
>
> > Q1 & Q3. EMV Priority Baseline Design and Comparison Protocol
>
> Thank you for your valuable comment.  We provide detailed clarifications as follows:
>
> - **Baseline Settings**: Our baselines consist of existing traffic signal control methods, each implemented with the same settings and hyperparameters as reported in their original works to ensure fair comparison.
> - **EMV Priority in Baselines**: All baselines incorporate a rule-based EMV priority(switching lights when EMV is detected under safe conditions), reflecting common practical strategies for emergency vehicle prioritization.
> - **Greedy EMV Baseline**: We follow the suggestion to add a rule-based baseline that switches the signal to green as soon as an EMV is detected and keeps it green until the EMV leaves. While this design minimizes EMV travel time, it significantly increases delay for regular vehicles, showing poor overall efficiency balance.
>
> | Method              | grid $T_{emv}$ | grid $T_{ve}$ | avenue $T_{emv}$ | avenue $T_{ve}$ | cologne $T_{emv}$ | cologne $T_{ve}$ | Fenglin $T_{emv}$ | Fenglin $T_{ve}$ |
> | ------------------- | -------------- | ------------- | ---------------- | --------------- | ----------------- | ---------------- | ----------------- | ---------------- |
> | Greedy EMV Baseline | 104            | 207.14        | 117              | 664.86          | 45                | 117.06           | 24                | 261.93           |
>
> > Q4. How to handle contentions among many simultaneous emergencies?
>
> Thank you for the insightful question.
>
> - Our method encodes the relation between each agent and multiple EMVs using a heterogeneous graph, and learns role embeddings via mutual information maximization. This ensures that agent roles reflect varying importance to different EMVs based on their spatial positions. For example, if an agent is spatially closer to EMV A than EMV B, its role embedding will more strongly reflect EMV A’s features, guiding behavior accordingly. We further design a role-similarity-based reward redistribution mechanism, enabling soft prioritization when multiple EMVs coexist.
>
> - We tested our model in scenarios with multiple concurrent EMVs, as reported in the main paper (Table 2). The results demonstrate that our method maintains high EMV efficiency while preventing excessive delay for REVs.
>
> [1]Wei, Hua, et al. "Colight: Learning network-level cooperation for traffic signal control." *Proceedings of the 28th ACM international conference on information and knowledge management*. 2019.
>
> [2]Wei, Hua, et al. "Presslight: Learning max pressure control to coordinate traffic signals in arterial network." *Proceedings of the 25th ACM SIGKDD international conference on knowledge discovery & data mining*. 2019.
>
> [3]Jiang, Haoyuan, et al. "X-Light: cross-city traffic signal control using transformer on transformer as meta multi-agent reinforcement learner." *Proceedings of the Thirty-Third International Joint Conference on Artificial Intelligence*. 2024.
>
> [4]Li, Zhenyu, Tianyi Shang, and Pengjie Xu. "Multi-Modal Attention Perception for Intelligent Vehicle Navigation Using Deep Reinforcement Learning." *IEEE Transactions on Intelligent Transportation Systems* (2025).
>
> [5]Jiang, Qize, et al. "Dynamic lane traffic signal control with group attention and multi-timescale reinforcement learning." IJCAI, 2021.
>
> [6]Wang, Tonghan, et al. "ROMA: multi-agent reinforcement learning with emergent roles." *Proceedings of the 37th International Conference on Machine Learning*. 2020.
>
> [7]Wang, Tonghan, et al. "RODE: Learning Roles to Decompose Multi-Agent Tasks." *International Conference on Learning Representations*.
>
> [8]Hu, Zican, et al. "Attention-Guided Contrastive Role Representations for Multi-agent Reinforcement Learning." *The Twelfth International Conference on Learning Representations*.

---

> ### Comment · Reviewer_Q23t · 2025-08-05
>
> Hi authors, thank you for your rebuttal. I think there are two remaining concerns:
>
> 1. W2 Unclear evaluation protocol is not addressed by the rebuttal
>
> >Datasets. Our datasets include two synthetic traffic scenarios, Grid4×4 and Avenue, as well as208two real-world scenarios, Cologne8 and FengLin [10, 11]. The synthetic scenarios are constructed209with idealized grid layouts containing uniformly distributed four-arm intersections. The real-world210scenarios feature a mix of two-arm, three-arm, and four-arm intersections. To simulate realistic211emergency conditions, we schedule the EMVs to depart from the middle point of the traffic episode.212Details about these datasets are shown in Appendix C.1.
>
> You mentioned that you used a dataset, although I understand the general reinforcement learning protol, I wonder how these dataset were used to split into train&test
>
>
> 2. Novelty
>
> I understand the MI role-based MARL has existed for a while, but the entirely missing the related work section on the specific topic makes your experiment design incomplete without situating your work in the previous work and setting yourself apart from the prior work.
> Instantiating the problem with a real-world problem is OK, but it should be considered an application paper instead of framing it as a new contribution to MARL.

---

> > ### Author Response · Authors · 2025-08-06
> >
> > > 1. W2 Unclear evaluation protocol is not addressed by the rebuttal
> >
> > Thank you for your comment. We would like to clarify that our method follows the standard online reinforcement learning setting, which differs fundamentally from supervised and offline learning paradigms. In supervised or offline learning, models are trained and evaluated on fixed datasets with explicit train/test splits. In contrast, online RL does not rely on static data. Instead, the agent collects trajectories through direct environment interaction during training, and its performance is evaluated by running the learned policy without further updates.
> >
> > In our work, the so-called “Grid”, “Cologne”, and other scenarios refer to simulation-based traffic environments rather than pre-collected datasets. These environments define road networks, signal layouts, and vehicle generation rules, but they do not contain fixed samples. The agent interacts with these environments over time, collecting trajectories and updating the policy accordingly.
> >
> > Evaluation is performed by running the learned policy—without further updates—on new episodes initialized with different random seeds. These seeds affect the traffic conditions (e.g., vehicle departure times and destinations), while the overall network structure remains the same. Therefore, there is no need for a traditional training/testing data split, as learning and evaluation are both conducted through environment interaction.
> >
> > > 2. Novelty
> >
> > Thank you for your insightful comment. We acknowledge that our original submission lacks a discussion on role-based reinforcement learning methods in the related work section. We will update the final version of the paper to include relevant literature on mutual-information-based role learning, such as ROMA[1], RODE[2], and ACORM[3].
> >
> > To further facilitate discussion, we conducted comparative experiments and found that directly applying existing role-based MARL methods to our signal-vehicle coordination task leads to suboptimal performance. We believe this is due to the following two limitations of prior approaches:
> >
> > - **Lack of consideration for heterogeneous agent interaction:** Existing methods are primarily designed for homogeneous agents and do not capture the spatial-temporal influence between different types of agents.
> >
> > - **No modeling of agent priority:** These approaches typically assume uniform importance across agents and fail to account for task-specific priority differences (e.g., emergency vehicles vs. regular traffic participants).
> >
> > | Method | grid $T_{emv}$ | grid $T_{ve}$ | avenue $T_{emv}$ | avenue $T_{ve}$ | cologne $T_{emv}$ | cologne $T_{ve}$ | Fenglin $T_{emv}$ | Fenglin $T_{ve}$ |
> > | ------ | -------------- | ------------- | ---------------- | --------------- | ----------------- | ---------------- | ----------------- | ---------------- |
> > | ACORM  | 194            | 228.80        | 501              | 674.65          | 213               | 230.86           | 168               | 254.35           |
> >
> > To address these challenges, our method introduces two key innovations:
> >
> > - **Heterogeneous graph-based role learning:** We incorporate a heterogeneous spatiotemporal graph and design three role-specific loss functions that encode both the spatial and temporal influence among different types of agents.
> >
> > - **Role-aware intrinsic reward:** We propose an intrinsic reward mechanism that assigns differentiated rewards based on role relevance. This encourages the model to prioritize behaviors that benefit high-priority agents, such as emergency vehicle passage.
> >
> > In summary, our work builds upon a real-world scenario to extend role-based MARL methods for heterogeneous, priority-aware agent coordination.
> >
> > [1]Wang, Tonghan, et al. "ROMA: multi-agent reinforcement learning with emergent roles." *Proceedings of the 37th International Conference on Machine Learning*. 2020.
> >
> > [2]Wang, Tonghan, et al. "RODE: Learning Roles to Decompose Multi-Agent Tasks." *International Conference on Learning Representations*.
> >
> > [3]Hu, Zican, et al. "Attention-Guided Contrastive Role Representations for Multi-agent Reinforcement Learning." *The Twelfth International Conference on Learning Representations*.

---

> > > ### Author Response · Authors · 2025-08-09
> > >
> > > We would like to politely follow up on our earlier response to your valuable comments.
> > >
> > > We hope our previous explanation has clearly conveyed our ideas and would like to sincerely confirm whether it has fully addressed your concerns. Should you need any further clarification, we are always ready to provide it.
> > >
> > > Thank you once again for taking the time to review our work and for your valuable guidance.

---

### Note · Authors · 2025-08-11

We sincerely thank the AC and all reviewers for their thoughtful feedback and for engaging with our work throughout the review process. We have carefully addressed all concerns raised, including the additional questions from Reviewer Q23t, and have provided detailed clarifications and justifications in our responses.

If the reviewers find our clarifications satisfactory, we would be grateful if they could kindly reconsider their scores in light of the additional information provided. Given the borderline nature of our paper, we also hope the AC will take these clarifications into account in the final decision.

We truly appreciate your time, expertise, and valuable suggestions, which have helped us improve the clarity and completeness of our work.

---

### Decision · Program_Chairs · 2025-09-17

**Decision:**

Accept (poster)

**Comment:**

This paper studies the challenge of emergency traffic control and the limitations of existing models. The authors propose the RMTC framework, which uses HTTG, dynamic role learning, and role-aware multi-agent RL to coordinate traffic components. The authors and reviewers had thorough rebuttal discussions, and finally, the reviewers had not converged to accept or reject this paper. The score is right on the borderline. After reading the paper & discussions, I personally think this paper is acceptable for publication in NeurIPS this year.